# Intra-Ring Variations and Interrelationships for Selected Wood Anatomical and Physical Properties of *Thuja Occidentalis* L. †

**Besma Bouslimi, Ahmed Koubaa \* and Yves Bergeron**

Institut de Recherche sur les forêts, Université du Québec en Abitibi-Témiscamingue; Rouyn-Noranda, Québec, QC J9X 5E4, Canada; besma.bouslimi2@uqat.ca (B.B.); yves.bergeron@uqat.ca (Y.B.)
\* Correspondence: ahmed.koubaa@uqat.ca; Tel.: +1-819-761-0971 (ext. 2579)
† This manuscript is part of the Ph.D. thesis of the first author available Online at depositium.uqat.ca.

**Abstract:** Intra-ring variation in wood density and tracheid anatomical properties and wood property interrelationships were investigated in *Thuja occidentalis* L. Samples were taken from three stands in Abitibi–Témiscamingue, Quebec, Canada. The structure of *T. occidentalis* wood is simple, homogeneous and uniform, which is desirable for wooden structures that require wood uniformity. From early- to latewood, cell and lumen diameter decreased, while cell wall thickness increased. These changes led to an increase of the cell wall proportion. Wood ring density and width interrelationships were weaker in mature wood compared to juvenile wood. Earlywood density is the more important in determining mature wood density than latewood density and proportion. Earlywood density explains 92% and 89% of the variation in juvenile and mature wood density, respectively. The negative relationship between ring density and width, although significant, was low and tends to weaken with increasing tree age, thus providing the opportunity for silvicultural practices to improve both growth and wood density. Ring width was positively and strongly correlated to early- and latewood width, but negatively correlated to tracheid length and latewood proportion. Accordingly, increases in ring width produce smaller tracheids and wider earlywood without a corresponding increase in latewood. Practical implications of the results are discussed.

**Keywords:** wood properties; ring density; anatomical properties; intra-ring variation; wood properties interrelationships

## 1. Introduction

*Thuja occidentalis* L., one of only two-arborvitae species native to North America, is distributed over a vast territory extending from the Gulf of St. Lawrence in the east to southeastern Manitoba in the west and from southern James Bay in the north to the Lake States in the south [1]. The timber of this species has a natural durability that enhances its utility in wooden structures exposed to constant moisture [2]. For example, the average service life of an untreated *T. occidentalis* heartwood post is 27 years, compared to just five years for an untreated black spruce post [3]. Hence, products such as shakes, shingles, fence posts and mulch made from *T. occidentalis* have considerable potential market value [4]. The low density and the excellent dimensional stability of this timber also increases its value for indoor joinery, outdoor furniture, and patios [5] making this species a particularly interesting opportunity for study. This work is an extension of a series of studies on wood quality variation in *T. occidentalis* [6–8].

Wood is a heterogeneous material, owing to its biological origin [9]. There are large variations in cell structure between and within trees and among species [10,11]. Within trees, the tracheid dimensions vary both with time of growth (earlywood/latewood) and with cambial age [12,13]. Cell parameters

such as length, wall thickness and width change systematically from early- to latewood and from the pith to the cambium of a wood trunk [14–17]. This wide variability in wood characteristics makes it difficult to assess accurately wood performance. Therefore, a better understanding of wood variability within a species would be useful for both wood quality research and utilization.

Wood density is an important wood quality attribute, owing to its high correlation with anatomical, physical and chemical properties. It is a measure of the total amount of solid-wood in a piece of wood [9] and depends on the proportions and dimensions of different cell types and spatial arrangements [10,18]. It can therefore be used to predict wood mechanical properties, pulp yield and paper quality [9,19,20]. Intra-ring wood density variation is indicative of wood uniformity and determines the suitability of a wood for specific products, especially for high-value-added applications [9,21]. For instance, uniform wood density is recommended for slicing and veneer peeling. For this application, it is important to know the magnitude and intra-ring wood density variation to determine the degree of wood uniformity, which is considered a limiting factor for transformation, especially during sawing, drying, machining, gluing and finishing [9].

The variation in intra-ring wood density is related to cambial activity and varies with age [18,22]. It also varies greatly from early- to latewood because of anatomical modifications [10,18]. The intra-ring variation depends on species and the type of the wood: juvenile and mature wood [21,23]. According to Koubaa et al. [24], the intra-ring density distribution in black spruce (*Picea mariana* (Mill) B.S.P (Britton-Sterns–Poggenburg)) is more homogeneous in juvenile wood, mainly due to lower latewood density and proportion. Anatomical changes are the main cause of wood density variation between early- and latewood, but variation in chemical composition has only a slight impact [10,18].

Juvenile wood is one of the most important sources of inter- and intra-tree wood variation, particularly in conifers [21,23]. Juvenile wood forms a central core around the pith from tree base to top, following the crown as it grows [9]. Typically, the properties of juvenile wood make a gradual transition toward those of mature wood [9,24]. The radial variation in wood ring density and width and tracheid size of *T. occidentalis* is greater in juvenile than in mature wood [6]. Compared to mature wood, juvenile wood is composed of smaller, shorter tracheids, higher wood ring density and latewood proportion, but lower ring width [6]. The longitudinal variation is more important in mature compared to juvenile wood [6]. Wood ring density and width decrease steadily from the tree base upward; however, tracheid length and width increase with increasing tree height [6].

It is also essential to determine wood ring density variation and its relationship with ring widths when assessing wood quality [25,26]. However, the correlation patterns between ring density and radial growth are complex. For example, earlywood and ring density were negatively correlated to ring and earlywood widths in black spruce, whereas ring density was positively correlated to latewood width in juvenile wood [27]. These correlations tend to weaken with increasing tree age. In balsam fir (*Abies balsamea*), the correlation between ring density and ring width was negative and significant in the butt log, however, it became not significant above a 3.0 m height [11]. It is therefore important to account for within-tree variation, which might partially explain the relationship between wood ring density and width. The radial and axial variations in ring width, ring density and tracheid length and width in *T. occidentalis* were investigated in a previous report [6]. However, no study to date has examined the relationship between ring density components and ring widths in this species. The variation in the relationship between ring width and tracheid length and width is also unknown. Furthermore, the intra-ring variations of wood density and tracheid anatomical properties (cell and lumen diameter, perimeter and area as well as cell wall thickness and proportion) in this species have not been studied to date. Thus, the main objective of this study was to investigate intra-ring variations of wood density and tracheid anatomical properties and wood property interrelationships in *T. occidentalis* grown in the Abitibi–Témiscamingue region of Québec, Canada. The specific objectives were to (1) determine intra-ring wood density variation as related to cambial age; (2) examine the variation in tracheid anatomical properties between early-and latewood; (3) investigate the interrelationships among ring widths and ring density components, and (4) examine the relationship between ring width and tracheid length and width.

## 2. Materials and Methods

### 2.1. Study Materials

The study materials and samples were described in a previous report [6]. Trees were sampled from three mature *T. occidentalis* stands in the Abitibi-Témiscamingue region of the province of Quebec, Canada: (Abitibi (48°28′ N, 79°27′ W); Lac Duparquet (48°25′ N, 79°24′ W), and Témiscamingue (47°25′ N, 78°40′ W)). Stand, tree and soil characteristics were detailed in previous reports [6,28]. In all, 45 trees (15 per site) were randomly sampled from the selected sites. The sampled trees were felled and total height, crown height, and diameter at breast height (DBH) were measured using a steel tape. The trees had similar heights and DBHs at the three stands, averaging 11 to 12 m height and 30 cm DBH at the Abitibi and Lac Duparquet sites and 27 cm DBH at the Témiscamingue site [6]. The age of sampled trees ranged from 60 to 198 years, with an average of 96 (60–134), 121 (73–198), and 93 (75–127) years for the Abitibi, Lac Duparquet and Témiscamingue sites, respectively. From each felled tree, 10 cm-thick disks were sampled at DBH. Disks were placed on pallets and air-dried with fans for several months to avoid decomposition until sample measurement.

### 2.2. Wood Ring Density and Width Measurement

Thin strips (20 mm wide and 1.57 mm thick) were sawn from each disk (bark to bark passing through the pith) and then extracted [6] to remove extraneous compounds [29]. Samples were then air-dried under restraint to prevent warping and conditioned to 8% equilibrium moisture content before measurement.

Ring density components were measured using a QTRS-01X Tree-Ring X-Ray Scanner (Quintek Measurement Systems, Knoxville, TN, USA), with a linear resolution step size of 20 µm. The mass attenuation coefficient ($cm^2 \ g^{-1}$) required to calculate the density was determined using the maximum moisture content method [6,30]. During scanning, precautions were taken to eliminate incomplete or false rings and rings with compression wood or branch traces. From the wood density profiles, annual (RD), earlywood (EWD) and latewood (LWD) density and annual (RW), earlywood (EWW) and latewood (LWW) ring width were determined. The boundary between early- and latewood was delineated using the maximum derivative method [21]. Latewood proportion (LP) was calculated as the ratio of LWW to RW. False and missing rings were detected by cross-dating ring width chronologies, using the COFECHA software [31]. One strip was discarded because of decay. Accordingly, a total of 44 strips were analyzed by X-ray densitometry.

The pith-to-bark variation in wood traits is frequently described in terms of juvenile and mature wood zones and is used to estimate the transition age. According to Bouslimi et al. [6], the radial variation profile of tracheid length increased steadily with ring number from pith outwards to stabilize at cambial age 30 in *T. occidentalis*. Considering the pattern of variation of tracheid length and that of other wood density traits, such as RD, RW, and LP [6], the wood produced from the pith up to the 30th ring is considered to be juvenile and the remaining mature wood.

Matlab software was used to model the intra-ring wood density profiles, using the maximum derivative method [21]. The intra-ring wood density variation was calculated for the whole tree and juvenile (rings 2–30) and mature wood (rings 31–100) (Figure 1a). The intra-ring wood density variation was also considered for selected ring groups (rings 2–10, rings 11–20 and every 10th annual ring up to the bark) (Figure 1b) to investigate the cambial age effect on this variation.

### 2.3. Tracheid Length and Width Measurement

Ten decay-free trees were randomly selected from the studied sites for tracheid length and width analysis. A 1 cm long and 1 cm deep pith-to-bark plank was sawn from each disk and prepared for analysis. Whole ring thin longitudinal specimens were extracted from each disk at systematic cambial ages (5, 10, 12, 15, 20, 25, 30 and every 10th annual ring up to the bark) and were then macerated using Franklin's [32] method. Each specimen was placed in a test tube, immersed in the Franklin solution

(1:1 (v/v) hydrogen peroxide diluted to 30% and concentrated glacial acetic acid), and kept in hot distilled water (90 °C) for seven hours until complete lignin dissolution. The delignified specimen was gently shaken in water with a laboratory blender to obtain a tracheid suspension. Tracheid length and width were measured automatically using a Fiber Quality Analyzer (FQA, OPTEST Equipment Inc., Hawkesbury, ON, Canada) [33]. A total of 5000 fibers were measured for each sample. Individual tracheid length (weight-weighted mean fiber length) and width are reported to a precision of 0.01 mm and 1 μm, respectively [33].

*2.4. Microscopic Analysis of Tracheid Anatomical Properties*

Samples for the anatomical analysis were cut from mature trees and studied using both light microscopy (LM) and scanning electron microscopy (SEM). Transverse cross sections of 20–25 μm thickness were cut from 1 cm wide wood samples with a sliding microtome (HM 325 Rotary Microtome, Microm International GmbH, Walldorf, Germany). Sections were then stained with safranin (1% in water) to enhance the contrast between cell wall and lumina. Sections were then placed on slides and fixed with cover slips. Using a camera (AxioCam MRc, Carl Zeizz AG, Carl Zeiss MicroImaging GmbH, Göttingen, Germany) installed on a LM (Axiovert25, Carl Zeizz AG, Carl Zeiss MicroImaging GmbH, Göttingen, Germany), the transverse sections were converted into image files to measure various tracheid anatomical properties: lumen length (radial lumen diameter) (μm), width (tangential lumen diameter) (μm), area (μm$^2$) and perimeter (μm), cell total length (radial cell diameter) (μm), width (tangential cell diameter) (μm), area (μm$^2$) and perimeter (μm); and cell wall thickness (μm) using WincellTM software [34]. The length and the width were measured as the horizontal and the vertical size of the cell in its center of gravity position [34]. Therefore, the length and width correspond to the radial and tangential cell size. Cell wall thickness was measured in the radial direction [34]. Cell wall area was calculated as the difference between the cell total area and lumen area. Lumen and cell wall proportions were also calculated as the ratio of lumen area and cell wall area to cell total area, respectively. Finally, the ratio between radial and tangential diameter was determined (R/T). All measurements were treated as time series. The attribution of tracheids to early- or latewood was determined using WincellTM software, following Mork's formula (1928) as explained by Denne [35]. The latewood cell is a cell for which twice the double wall thickness is greater than or equal to the lumen length [35].

Small specimens (2.5 by 2.5 by 1 cm) were also cut from sampled disks and transverse, radial and tangential cross-sections were prepared for SEM testing. Specimens were first softened by overnight soaking in water and then oven-dried for two hours at 100 °C. The area of interest was cut out with a razor and platinum deposits followed by carbon deposits were applied to the surface. The surfaces were then observed using a SEM (JSM-840A, SRNEML, Oklahoma City, OK, USA).

*2.5. Statistical Analysis*

Wood density was subjected to a variance analysis (ANOVA) using a mixed-model approach [36] where ring groups, ring width and site were considered fixed effects and tree a random effect. The hierarchical effects of individual tree and site were accounted for using two nested levels, with the tree effect nested within the site effect, as follows:

$$Y_{ijkl} = \mu + \alpha_i + \beta^j + \gamma_k + \delta_l + \varepsilon, \tag{1}$$

where $Y$ is the dependent variable, $\mu$ the grand mean, $\alpha_i$ is the fixed effect of ring groups, $\beta_j$ is the fixed effect of ring width, $\gamma_k$ is the fixed site effect, $\delta_l$ is the random tree effect, and $\varepsilon$ is the residual error. Z-tests results must be considered as indicative only, given that there were only three site replicates [36]. A total of 38,500 growth rings were studied and ten systematic ring groups (rings 2–10, rings 11–20, rings 21–30, and every 10th annual ring up to ring 100) were retained for the analysis.

The SAS mixed-model procedure (PROC MIXED) was used to fit the models using restricted maximum likelihood (REML) [37]. Degrees of freedom were determined using the Kenward–Roger

method. The statistical significance of fixed effects was determined using F-tests at $p < 0.05$. Z-tests were conducted to determine whether the random effect significantly differed from zero [36]. Variance components were estimated as a percentage of total variation (VAR) of all effects, using the VARCOMP procedure. The mean, the coefficient of variation and the standard deviation for wood density and width were calculated for each selected ring group. Tukey's multiple range method was used to test significant statistical differences in wood density of selected ring groups Differences were considered statistically significant at $p < 0.05$.

Pearson's correlation coefficients were also computed using the CORR procedure to determine relationships between all wood density components in juvenile and mature wood, and for selected ring groups. Correlation analyses were also conducted between tree height, tree diameter, ring width and tracheid length and width. Tukey's multiple range method was also used to test significant statistical differences in anatomical properties between early-and latewood. All statistical and correlations analyses were performed using SAS (SAS, 2008, SAS Institure Inc., Cary, NC, USA) [37].

## 3. Results and Discussion

### 3.1. Intra-Ring Wood Density Variation and Anatomical Changes between Early-and Latewood

The intra-ring wood density variations of the whole tree and the juvenile and mature wood for 44 *T. occidentalis* trees are shown in Figure 1a. Intra-ring wood density variation is also examined for selected ring groups (Figure 1b). The cell structure of *T. occidentalis* wood is presented in Figure 2. The means and variations in early-and latewood tracheid anatomical properties in are shown in Table 1. The means and variations of wood density at selected ring groups are shown in Table 2.

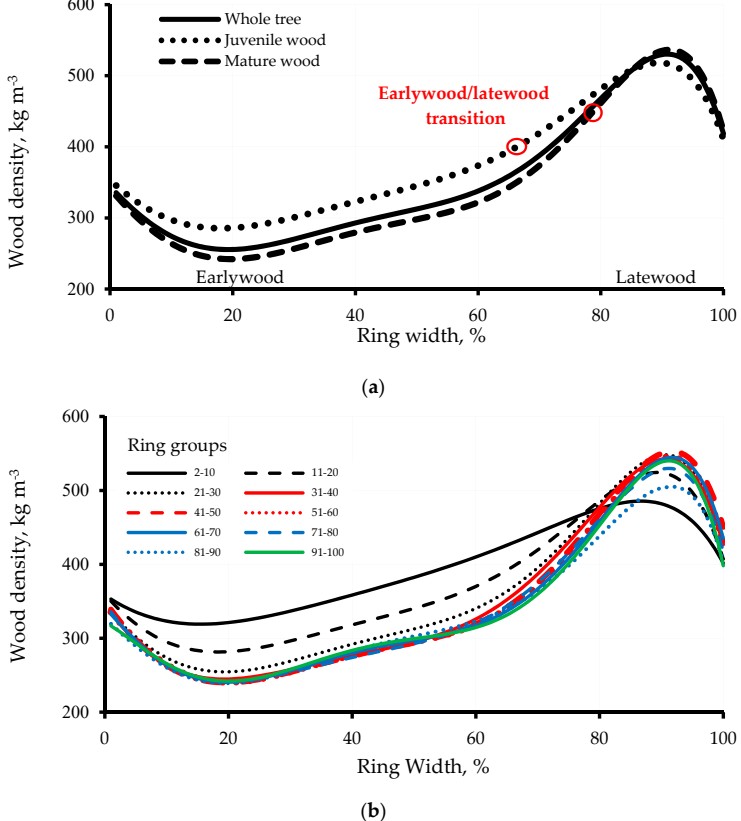

(a)

(b)

**Figure 1.** Intra-ring wood density profiles of: (**a**) the whole tree (rings 2–100 rings), the juvenile (rings 2–30) and mature wood (rings 31–100) and (**b**) selected ring groups in 44 T. *occidentalis* trees grown in the Abitibi–Témiscamingue region, Rouyn–Noranda, Québec, Canada.

The intra-ring wood density (Figure 1) and tracheid size variations (Figure 2 and Table 1) revealed that the transition from early- to latewood is gradual in *T. occidentalis*. The early- and latewood were distinguished in the structure (Figure 2 and Table 1). The latewood zone was narrow with thick cell walls (Figure 2a–c, Table 1) and was slightly denser than the earlywood zone (Figure 1a). *Thuja occidentalis* wood was relatively homogeneous and simple in structure, consisting primarily of overlapping tracheids (tracheids average 23.58 μm and 26.19 μm in the tangential and the radial diameter, respectively) connected by uniseriate xylem rays, ray parenchyma cells (Figure 2c, Table 1) and uniseriate, or occasionally biseriate, bordered pits (Figure 2d). Axial parenchyma was usually rare or absent. Rays are uniseriate and thin (Figure 2c). The tracheid cell walls were organized in layers of different thicknesses: thick latewood cell walls and thin earlywood cell walls (Figure 2a–c, Table 1). In earlywood, the longitudinal tracheids were hexagonal with minimal wall thickness and a larger diameter, usually in the radial direction, while the diameter of longitudinal tracheids in the tangential direction remained relatively constant within a growth ring (Figure 2a,c). In latewood, the cross sections of tracheids were essentially rectangular and compacted radially to a tabular shape (Figure 2a,c).

**Table 1.** Means and variations (standard deviation (SD) and coefficient of variance (CV)) in anatomical properties in early- and latewood *T. occidentalis* tracheids.

| Anatomical Properties | Earlywood (*n* = 392 cells) | | | Latewood (*n* = 366 cells) | | |
|---|---|---|---|---|---|---|
| | Means [1] | SD | CV | Means | SD | CV |
| Cell radial diameter (μm) | 26.19 [a] | 4.63 | 17.69 | 23.58 [b] | 3.97 | 16.84 |
| Cell tangential diameter (μm) | 23.59 [a] | 4.13 | 17.54 | 25.42 [b] | 3.71 | 14.58 |
| Cell area (μm$^2$) | 626.13 [a] | 180.70 | 28.86 | 608.75 [b] | 174.72 | 28.70 |
| Cell perimeter (μm) | 99.58 [a] | 14.80 | 14.87 | 97.99 [b] | 13.91 | 14.19 |
| Lumen radial diameter (μm) | 21.86 [a] | 4.49 | 20.53 | 12.97 [b] | 2.42 | 18.66 |
| Lumen tangential diameter (μm) | 19.25 [a] | 4.18 | 21.70 | 14.80 [b] | 2.70 | 18.25 |
| Lumen area (μm$^2$) | 375.21 [a] | 120.04 | 31.99 | 165.81 [b] | 48.37 | 29.17 |
| Lumen perimeter (μm) | 82.21 [a] | 14.53 | 17.67 | 55.55 [b] | 7.91 | 14.25 |
| Cell wall thickness (μm) | 4.34 [a] | 1.81 | 41.76 | 10.61 [b] | 3.35 | 31.55 |
| Cell wall area (μm$^2$) | 250.92 [a] | 99.67 | 39.72 | 442.94 [b] | 167.12 | 37.73 |
| Lumen (%) | 60.19 [a] | 11.61 | 17.88 | 28.53 [b] | 7.96 | 27.91 |
| Cell wall (%) | 39.78 [a] | 10.77 | 27.07 | 70.48 [b] | 7.96 | 11.13 |
| Lumen R/T [2] | 1.14 [a] | 0.34 | 28.50 | 0.90 [b] | 0.22 | 24.59 |
| Cell R/T [3] | 1.11 [a] | 0.45 | 30.13 | 0.93 [b] | 0.33 | 25.65 |

[1] Multiple comparison tests were performed using a Tukey–Kramer adjustment between early- and latewood tracheid-anatomical properties. Means followed by a different letter indicate significant differences between early- and latewood at $p < 0.05$. [2] The ratio between lumen radial and tangential diameter. [3] The ratio between cell radial and tangential diameter; SD: standard deviation; CV: coefficient of variation.

Cell and lumen diameters decreased while cell wall thickness increased from early- to latewood (Figure 2, Table 1). Lumen diameter reduction was even greater than the one seen for cell diameter (Table 1). Cell and lumen diameter changes along a growth ring were greater in the radial direction than in the tangential one (Figure 2a,c, Table 1). For example, the lumen radial diameter decreased by 40% from early-to latewood, while the lumen tangential diameter decreased by only 23% (Table 1). This could explain the decrease in the ratio between radial and tangential diameter (Table 1). These changes led to a decline in cell and lumen perimeters. However, lumen perimeter reduction was greater than that for cell perimeter. For instance, lumen perimeter decreased by 33% from early-to latewood compared to only 1.6% for cell perimeter (Table 1). By contrast, cell walls were thicker in the tangential direction than in the radial one (Figure 2a,c). Thus, changes in cell wall thickness, and cell and lumen diameters led to an increase in cell wall area and proportion (Table 1). However, it is important to note that variations in cell wall thickness were larger compared to those of cell size (Table 1).

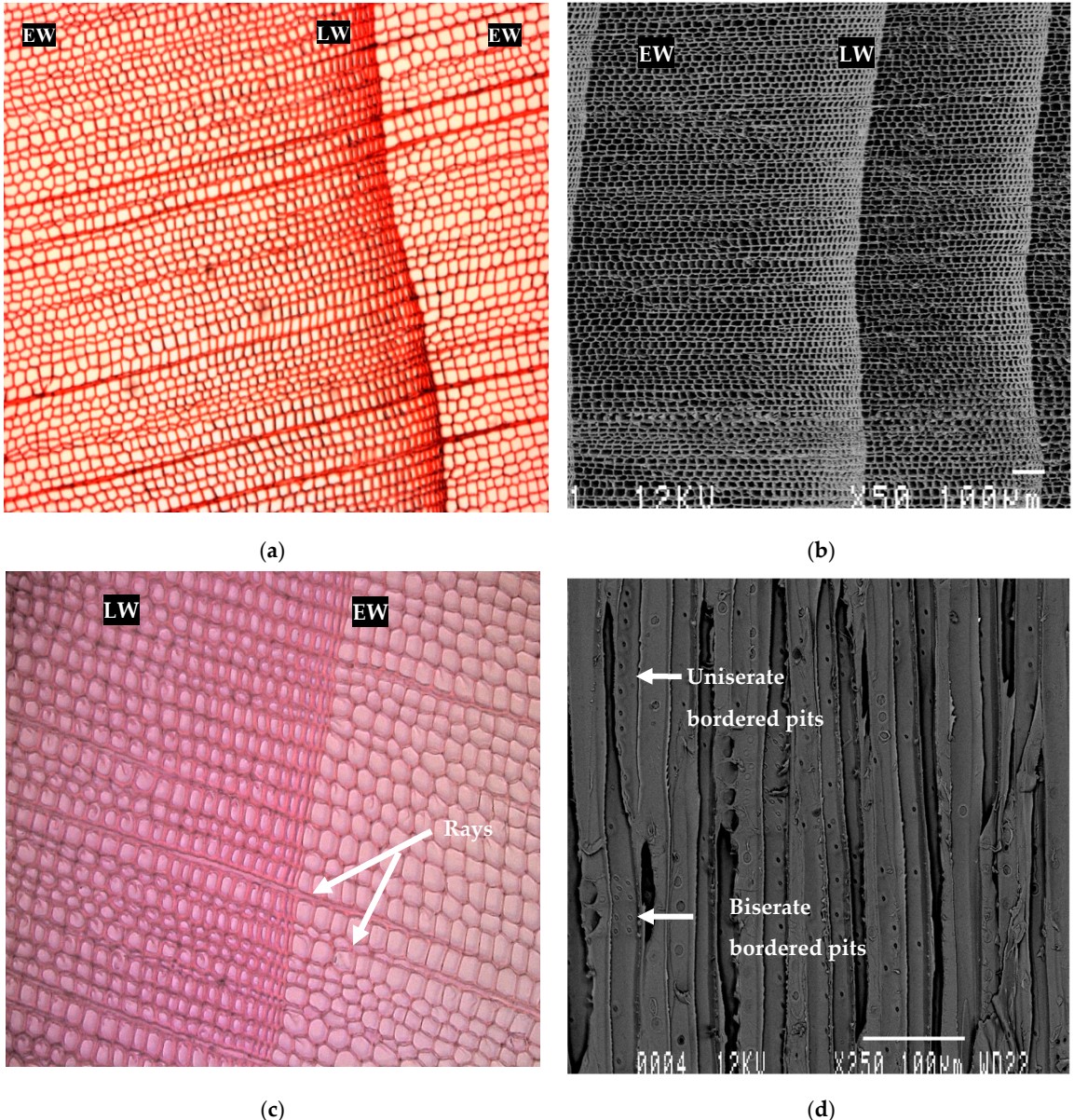

**Figure 2.** Cell structure of *T. occidentalis* wood: (**a,b**) transverse sections showing the gradual transition from early- to latewood (×10 (optical microscope) and ×50 (SEM), respectively); (**c**) transverse section (optical microscope) showing uniseriate xylem rays (×20) and (**d**) radial section (SEM) showing uniseriate and biseriate bordered pits in earlywood tracheids (×250). EW: earlywood, LW: latewood.

From early- to latewood, the tracheids became smaller, the cell and lumen areas decreased, whereas the cell wall area increased (Figure 2a,c, Table 1). For instance, the lumen area was double the size in early- compared to latewood (Table 1). However, cell wall area was more important in latewood, where it represented 71% of total cell area compared to only 39% in earlywood. This could explain the five-fold increase in the ratio between cell wall and lumen area from early- to latewood tracheids (Table 1). Similar findings were reported for other species such as Douglas fir (*Pseudotsuga menziesii* (Mirb.) Franco) [18,38], Norway spruce (*Picea abies*), Scots pine (*Pinus sylvestris*) and silver Fir (*Abies alba*) [10], as well as *Gmelina arborea* [39] and *Acacia mangium* plantations [40]. The cell size reduction (cell and lumen diameter) occurs mainly in the radial direction, while cell wall thickness increases occurs mainly in the tangential direction [18,39–42]. According to Hannrup et al. [41] and Rathgeber et al. [18], the cell and lumen tangential diameter reduction occurs mainly in latewood because anticlinal divisions occur more frequently at the end of the growing season. This may explain

the variation in cell and lumen diameter between early- and latewood in the radial and the tangential directions of *T. occidentalis* wood (Figure 2c and Table 1). Cell and lumen diameter changes were greater in the radial direction than in the tangential one. Wang et al. [43] already noted the cell wall thickness reduction at the end of a growth ring for black spruce (*Picea mariana* Mill.). However, it is important to note that the ratio of cell wall thickness to lumen diameter varied from 0.1 to 2.1 in black spruce compared to only 0.2 to 0.8 in *T. occidentalis* wood, this could explain the low intra-ring wood density variation of this wood (Figure 1).

The mean of wood density for the studied *T. occidentalis* trees (Table 2) was 355.41 kg m$^{-3}$, which is higher than that reported previously for this species in Wisconsin (281–324 kg m$^{-3}$) [44]. Wood density exhibited an S-shape profile from the beginning of a growth ring to its end, with a slight decrease occurring during the first 20% of a ring (Figure 1a). Consequently, wood density decreased by 18%, increasing thereafter by 106% to reach a maximum value (530 kg m$^{-3}$) at the 90% mark of the ring, before a slight decline (22%) at the end of the growth ring (410 kg m$^{-3}$). Moreover, LWD was about two times (41%) greater than EWD (Figure 1a). This variation is lower (260–530 kg m$^{-3}$) compared to that reported for other conifer species such as Norway spruce (351–1027 kg m$^{-3}$), Scots pine (325–939 kg m$^{-3}$) and silver fir (287–861 kg m$^{-3}$), where the LWD was about 3–4 times greater than the EWD [10,18,39]. The evolution of density was also more rapid in the latewood of *T. occidentalis* (Figure 1a). The earlywood represented more than 70% of the ring and the density variation with ring width was therefore more important in early- than in latewood. This result is in good agreement with previous reports on Norway spruce and Scots pine [10,18].

**Table 2.** Descriptive statistics (SD, CV, max and min) of wood density at selected ring groups in 44 *T. occidentalis* trees.

| Ring Groups [1] | Number | Means (kg m$^{-3}$) | SD (kg m$^{-3}$) | CV (%) | Max (kg m$^{-3}$) | Min (kg m$^{-3}$) |
|---|---|---|---|---|---|---|
| 2–10 | 4400 | 392.28 [a] | 105.65 | 26.93 | 762.07 | 171.13 |
| 11–20 | 4400 | 373.09 [b] | 102.35 | 27.43 | 766.59 | 194.57 |
| 21–30 | 4400 | 357.26 [c] | 110.02 | 30.80 | 733.92 | 191.91 |
| 31–40 | 4400 | 347.41 [d] | 108.89 | 31.34 | 732.23 | 195.72 |
| 41–50 | 4400 | 345.88 [d] | 113.43 | 32.79 | 854.73 | 200.02 |
| 51–60 | 4300 | 343.90 [d] | 110.03 | 32.00 | 687.39 | 194.20 |
| 61–70 | 4200 | 344.71 [d] | 107.35 | 31.14 | 669.32 | 191.98 |
| 71–80 | 3600 | 343.78 [d] | 110.91 | 32.26 | 715.60 | 194.75 |
| 81–90 | 2600 | 341.09 [d] | 110.09 | 32.28 | 676.99 | 185.76 |
| 91–100 | 1200 | 354.71 [e] | 118.45 | 33.39 | 704.61 | 198.02 |
| All data | 3,8500 | 354.41 | 109.72 | 31.04 | 854.74 | 171.13 |

[1] Multiple comparison tests of different ring groups were performed with Tukey–Kramer adjustment. Means followed by the same letter indicate no significant difference between ring groups at $p < 0.05$.

The patterns of intra-ring variation were relatively similar between rings from juvenile and mature wood (Figure 1a). However, the intra-ring distribution of latewood was slightly more important in juvenile (32%–40%) than in mature wood (20%) (Figure 1a). For instance, the latewood transition occurred at 65% of the ring for ring 6 (Figure A1). With increasing age, the earlywood/latewood transition zone occurred later in the ring, at 70% of the 20th ring width. In mature wood, the intra-ring distribution of LP was relatively uniform, accounting for 20% of the ring (Figure A1). The pattern of wood density variation in individual rings of *T. occidentalis* appears to be comparable to the general pattern reported for other conifers species, such as white spruce (*Picea glauca* (Moench) Voss), eastern hemlock (*Tsuga canadensis* (L.) Carrière), jack pine (*Pinus banksiana* Lamb.) [45] and Douglas fir [18]. However, *T. occidentalis* wood density is more uniform and homogeneous compared to these species and is characterized by a moderate difference between early- and latewood (270 kg m$^{-3}$). This is beneficial for use in wooden structures that require wood uniformity such as veneer peeling and slicing [46].

The analysis of variance (Table 3) showed that ring groups as well as ring width and tree significantly affected wood density. The site effect could be masked by the tree-to-tree variation [6].

**Table 3.** Linear mixed model analysis of variance, with F-values for fixed effects, Z-values for random effects, their significance and the variance component (VAR COMP) of each source of variation for wood density in *T. occidentalis*.

| Fixed Effects | | | | |
|---|---|---|---|---|
| **Sources** | **Df [1]** | **F-Value** | ***p*-Value** | **VAR COMP (%)** |
| Site | 2 | 1.48 | 0.24 | 0.11 |
| Ring groups | 10 | 331.33 | <0.0001 | 2.02 |
| Ring width | 99 | 943.24 | <0.0001 | 69.14 |
| **Random Effects** | | | | |
| **Sources** | **df** | **Z Value** | ***p*-Value** | **VAR COMP (%)** |
| Tree | 14 | 4.38 | <0.0001 | 0.28 |
| Residual | 137 | 140.62 | <0.0001 | 28.44 |

[1] Degree of freedom.

Ring width was the most important source of variation in wood density, accounting for 69.1% of the total variation, followed by ring groups (2.02%). Wood density variation is related to various factors including cambial age and ring width [6,9]. Hence, the effects of ring group and ring width on wood density were highly significant ($p < 0.001$) (Table 3). Overall, wood density decreased steadily with cambial age (Table 2). The wood density variation with cambial age was minimal near the pith, increased with increasing ring width up to 60% of the ring and decreased thereafter (Figure 1b).

The intra-ring wood density variation with cambial age showed substantial changes (Figure 1b). In earlywood, wood density decreased with cambial age. However, wood density showed the reverse pattern in latewood, increasing with cambial age. The intra-ring wood density distribution was more homogeneous in mature than in juvenile wood (Figure 1b, Table 2). This result disagrees with a previous report on black spruce, where this distribution was more homogeneous in juvenile wood, mainly due to lower LWD and LP by width [24]. According to Zobel and Van Buijtenen [9], the larger variation in intra-ring wood density within the first 30 rings (Figure 1) could be linked to the variation in juvenile properties. Compared to mature wood, juvenile wood is characterized by a higher variation in cell dimensions and cell wall formation [47,48]. In *T. occidentalis*, the LP (Figure 3) was higher near the pith (65%), decreasing gradually to reach a minimum in the transition zone (20%) and remained constant thereafter [6]. Accordingly, the intra-ring density distribution in this wood is more homogenous in the mature wood, which is mainly attributable to the lower LP by width compared to juvenile wood.

Wood density decreased progressively to about age 30 in earlywood and remained constant thereafter (Figure 1b). In contrast, LWD showed a different pattern: it increased from a minimum near the pith (rings 2–10) to reach a maximum at the juvenile-mature wood transition zone (rings 21–30). However, the variation in wood density with cambial age was minimal beyond the 30th ring from the pith, although a consistent slight decrease was observed with increasing cambial age thereafter. Furthermore, the variation in wood density with cambial age was more important in early- than in latewood (Figure 1b). The anatomical changes are primarily responsible for intra-ring wood density variation [10,18,49]. Earlywood lumen diameter and LP are the two most important predictors of wood density [18,41]. LP can explain up to 60% of the density variation [38]. Decoux et al. [10] propose that intra-ring variation of wood density is mainly due to the variation in cell wall thickness. Given the tubular shape of tracheids, cell wall thickness is of great importance when stiffness of the tracheids is compared. Earlywood cells have a thin wall layer, which is one of the reasons that earlywood is less dense compared to latewood [9].

For *T. occidentalis*, the latewood tracheids are characterized by narrow cell lumina and thick cell walls compared to earlywood (Figure 2, Table 1). From early- to latewood, cell wall proportion increased, while cell and lumen area decreased. Thus, the wood density increase in latewood (Figure 1) is mainly linked to cell wall thickness (Table 1). These results are in good agreement with the findings

of Rathgeber et al. [18] for Douglas fir. According to their results, cell wall area and thickness decreased slightly at the beginning and at the end of a growth ring. Cell wall thickness reduction at the end of a growth ring was already reported for black spruce [43], Norway spruce, Scots pine and silver fir [10]. This explains the steady decrease observed in wood density at the beginning and the end of the ring in *T. occidentalis* (Figure 1). Deleuze and Houiller's process-based model of xylem growth simulated a reduction in wood density at the beginning of a ring and indicated the reduction was caused by a temporary substrate shortage used for cell wall construction [50]. Wimmer [51] showed that radial diameter and cell wall thickness of latewood tracheids have the greatest influence on the growth ring density of conifers, while tracheid length and other cell characteristics, such as parenchyma rays and resin canals, can be neglected. For hardwoods such as green ash (*Fraxinus pennsylvanica* Marsh) [52] and red maple (*Acer rubrum* L.) [53], the higher LWD was attributed to an increase in fiber cell wall thickness and a decrease in vessel diameter compared to those of earlywood.

The variation of intra-ring wood density with cambial age (Figure 1b) is mainly associated with anatomical changes, which are related to cambial activity [18,41]. Hence, tracheid dimension varies with tree age [48,54]. Hannrup et al. [41] reported that the total radial and tangential lumen diameter decreased with the age of the wood, which could explain the increase in the proportion of the narrow, thick-walled latewood tracheid. The radial and tangential lumen diameter of the earlywood increased with the age of the wood, which agrees with the age trend normally occurring in conifers [16,55]. The tangential lumen diameter of the studied trees was consistently smaller in late- than in earlywood (Table 1). This may be the result of thicker cell walls in the latewood, but it may also be because anticlinal cell divisions are limited to the latewood [55]. This may explain the increase in LWD with increasing cambial age in *T. occidentalis* (Figure 1b).

## 3.2. Interrelationships of Wood Ring Density and Width

Table 4 shows the correlation coefficients for ring density components and ring widths in juvenile and mature wood. The relationships between wood ring density and width traits in *T. occidentalis* are also examined for selected ring groups (Table 5). Overall, the correlations between ring density components were statistically significant ($p < 0.01$) (Table 4). In juvenile wood, the correlation between RD and EWD was positive and stronger than that between RD and LWD and LP. However, in mature wood, the RD-LWD correlation was weaker. This indicates that the magnitude of the correlation differs between juvenile and mature wood in *T. occidentalis*. Similar results were reported for black spruce [27] and balsam fir [26].

**Table 4.** Pearson correlation coefficients between ring density components and ring widths in juvenile and mature wood of *T. occidentalis* and their statistical significance.

| Wood Density Properties | | RD | EWD | LWD | RW | EWW | LWW | LP |
|---|---|---|---|---|---|---|---|---|
| | | | | | **Juvenile Wood** | | | |
| | RD | 1 | 0.92 *** | 0.53 *** | −0.1 ** | −0.15 ** | 0.05 ns | 0.23 *** |
| | EWD | 0.89 *** | 1 | 0.31 *** | −0.1 ** | −0.11 ** | −0.02 ns | 0.15 ** |
| | LWD | 0.37 *** | 0.25 *** | 1 | 0.13 ** | 0.21 *** | −0.1 ** | −0.31 *** |
| Mature wood | RW | −0.19 ** | −0.03 ns | 0.11 ** | 1 | 0.94 *** | 0.66 *** | −0.34 *** |
| | EWW | −0.23 *** | −0.02 ns | 0.26 *** | 0.96 *** | 1 | 0.36 *** | −0.58 *** |
| | LWW | −0.03 ns | −0.017 ns | −0.34 *** | 0.59 *** | 0.32 *** | 1 | 0.36 *** |
| | LP | 0.28 *** | 0.11 ** | −0.63 *** | −0.38 *** | −0.59 *** | 0.40 *** | 1 |

Significance level: * = $p < 0.05$, ** = $p < 0.01$, *** = $p < 0.001$ and ns = not significant; RD: annual ring density; EWD: earlywood density; LWD: latewood density; TD: transition density; RW: annual ring width; EWW: earlywood width; LWW: latewood width; LP: latewood proportion.

For *T. occidentalis*, the correlation between RD and LP was weak compared to correlations between RD and EWD and LWD (Table 4). Thus, EWD and LWD are more important in determining RD than LP in this species. EWD explained 92% and 89% of the variation in juvenile and mature wood density, respectively. This result is somewhat different form previous findings on black spruce [27,56], where LP ($r = 0.91$) was more important than EWD ($r = 0.86$) and LWD ($r = 0.59$) in determining RD. Koga

and Zhang [26] also reported that EWD ($r = 0.83$) and LP ($r = 0.40$) were the most important parameters in determining RD of balsam fir. The percentage of early-and latewood in a growth ring determines the overall density of the ring. In *T. occidentalis*, LWW is relatively constant over age; therefore, RD is closely related to EWD (Figure 3) [6]. Moreover, the correlation between RD and EWD was nearly constant (about 0.90) over tree age while the correlations between RD and LWD decreased consistently with cambial age (Table 5). Thus, EWD is the most important parameter in determining mature wood density in *T. occidentalis*.

The correlation between EWD and LWD was positive in both juvenile and mature wood (Table 4). This result contradicts previous findings for Douglas-fir where the correlation was negative in both juvenile and mature wood [57]. Zhang et al. [58] also reported a negative correlation in the juvenile wood of black spruce. No significant correlations between EWD and LWD were found in the mature wood of black spruce [27] and balsam fir [26].

RD and EWD were positively correlated to LP, whereas LWD was negatively correlated to LP (Table 4). This result disagrees with previous findings on black spruce [27], Douglas fir [57] and balsam fir [26], where the correlation between LP and LWD was positive. The negative correlation between LWD and LP in *T. occidentalis* may be explained by their patterns of radial variation (Figure 3). LWD was low near the pith, increased to a maximum in the juvenile-mature transition zone (30 years) and remained constant thereafter. In contrast, LP was higher near the pith, decreased gradually to reach a minimum in the transition zone and remained constant afterward [6]. Hence, the relationship between LWD and LP was negative and weaker in juvenile wood, but stronger and almost constant in mature wood (Tables 4 and 5).

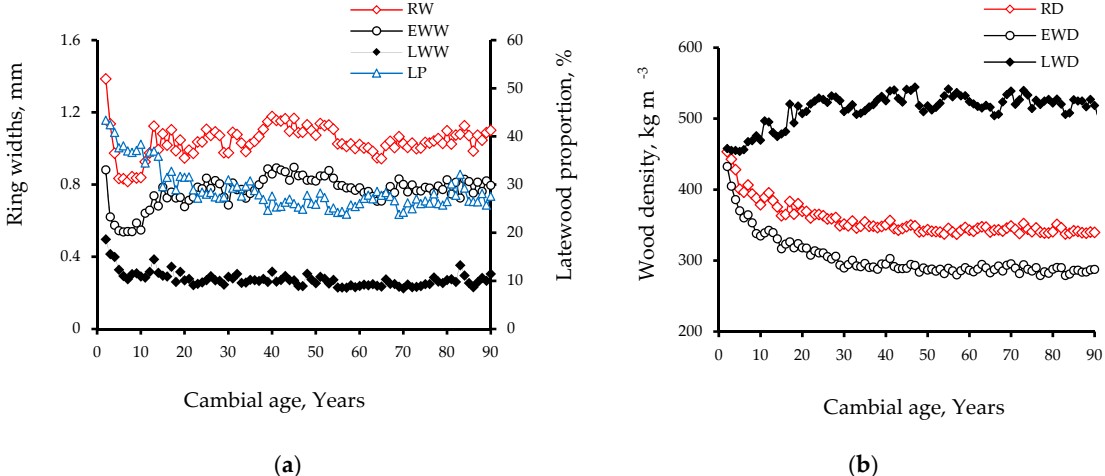

(**a**)

(**b**)

**Figure 3.** Radial variation patterns for: (**a**) annual ring width (RW, mm), earlywood width (EWW, mm), latewood width (LWW, mm) and latewood proportion (LP, mm) and (**b**) annual ring density (RD, kg m$^{-3}$), earlywood density (EWD, kg m$^{-3}$) and latewood density (LWD, kg m$^{-3}$) in 44 *T. occidentalis* trees.

This study also revealed a negative relationship between the ring width and ring density components of *T. occidentalis* (Table 4). RD correlated significantly and negatively with RW and EWW in juvenile and mature wood, but no significant correlation between RD and LWW was found (Table 4). The correlations between RD and both RW and EWW were weaker in mature wood (Table 4). This suggests that the negative impact of high growth on density decreases when the wood reaches maturity.

The effects of ring widths on ring density components were also examined for selected ring groups (Table 5). The negative relationships between RW and RD, although significant, were low ($r < -0.1$) and tended to weaken slightly with increasing cambial age. These results are in good agreement with previous findings for black spruce [27] and balsam fir [26]. Using all the sample data available for RW and RD, results also show a weak relationship between the two variables (Figure 4a). The coefficient of determination between RW and RD was low but significant. According to Koga and Zhang [26], a



slow increase in RW could negatively affect RD. The percentage of early-and latewood in a growth ring determines the overall density of the ring, and in *T. occidentalis*, the LWW is relatively constant. Therefore, the RW is closely related to the EWW (Figure 3). As the RW increases, the width of the earlywood increases without a corresponding increase for latewood, causing lower RD [6]. This could explain the not significant correlation between RD and LWW in both juvenile and mature wood, as well as the negative relationship of RD with LP (Tables 4 and 5). In juvenile wood, earlywood cells have a thin wall (Table 1), which explains its lower density [9]. According to Zhang et al. [59], the decrease in RD combined with the increase in RW in conifers was more pronounced in species that showed a gradual transition from early- to latewood (as *T. occidentalis*) than in species with an abrupt transition.

**Table 5.** Pearson correlation coefficients between ring density components and ring widths for selected ring groups in *T. occidentalis* and their statistical significance.

| | Ring Groups | | | | | | | | |
|---|---|---|---|---|---|---|---|---|---|
| | 2–10 | 11–20 | 21–30 | 31–40 | 41–50 | 51–60 | 61–70 | 71–80 | 81–90 |
| RD-EWD | 0.94 *** | 0.89 *** | 0.86 *** | 0.81 *** | 0.84 *** | 0.90 *** | 0.87 *** | 0.91 *** | 0.92 *** |
| RD-LWD | 0.81 *** | 0.57 *** | 0.21 *** | 0.43 *** | 0.36 *** | 0.25 *** | 0.27 *** | 0.33 *** | 0.14 * |
| EWD-LWD | 0.61 *** | 0.30 *** | 0.01 $^{ns}$ | 0.32 *** | 0.30 *** | 0.19 ** | 0.23 *** | 0.26 *** | 0.07 $^{ns}$ |
| RD-LP | 0.07 $^{ns}$ | 0.26 *** | 0.39 *** | 0.16 ** | 0.31 *** | 0.47 *** | 0.47 *** | 0.40 *** | 0.43 *** |
| LWD-LP | −0.05 $^{ns}$ | −0.22 *** | −0.57 *** | −0.63 *** | −0.61 *** | −0.55 *** | −0.53 *** | −0.57 *** | −0.68 *** |
| EWD-LP | −0.03 $^{ns}$ | 0.14 ** | 0.21 *** | −0.12 * | 0.02 $^{ns}$ | 0.27 *** | 0.2 *** | 0.24 *** | 0.32 *** |
| RD-RW | −0.13 * | −0.06 $^{ns}$ | −0.19 ** | −0.12 * | −0.12 * | −0.13 ** | −0.19 ** | −0.36 *** | −0.32 *** |
| EWD-RW | −0.10 $^{ns}$ | 0.07 $^{ns}$ | −0.07 $^{ns}$ | 0.17 ** | 0.11 * | 0.10 * | −0.02 $^{ns}$ | −0.15 ** | −0.07 $^{ns}$ |
| LWD-RW | −0.06 $^{ns}$ | 0.15 ** | 0.13 * | 0.11 * | 0.25 *** | 0.11 * | 0.21 *** | −0.08 $^{ns}$ | −0.12 * |
| LP-RW | −0.29 *** | −0.28 *** | −0.42 *** | −0.39 *** | −0.52 *** | −0.39 *** | −0.46 *** | −0.36 *** | −0.29 *** |

Significance level: * = $p < 0.05$, ** = $p < 0.01$, *** = $p < 0.001$ and ns = not significant; RD: annual ring density; EWD: earlywood density; LWD: latewood density; RW: annual ring width; LP: latewood proportion.

The correlation between EWD and RW and EWW was negative in juvenile wood and not significant in mature wood (Table 4). These results contradict previous findings for black spruce [27,58–60] and balsam fir [11], where the correlations were significant in both juvenile and mature wood, and only in mature wood [26], respectively. For selected ring groups, the correlations between EWD and RW over tree age were weak or not significant (Table 5). However, the correlations between both RW and EWW and LWD were positive and significant in both juvenile and mature wood (Table 4). The correlation increased with cambial age (Table 5). This result contradicts previous findings for black spruce [27], where LWD correlated negatively to RW ($r = −29$) and EWW ($r = −41$) in mature wood. For balsam fir [11], LWD was correlated to RW in mature wood only ($r = 0.50$).

LP correlated negatively to RW in both juvenile and mature wood (Table 4). These correlations were moderate (Figure 4b). The negative relationship tended to increase with cambial age (Table 5). For instance, it was −0.29 for rings 2 to 10 compared to −0.46 for rings 61 to 70. According to Bouslimi et al. [6], RW is closely related to EWW in *T. occidentalis* because the LWW is constant over tree age. Therefore, LP decreased with cambial age because the increasing RW produced wider earlywood without a corresponding increase in LWW (Figure 3). Accordingly, the correlation between RW and EWW was very strong compared to that between RW and LWW in both juvenile and mature wood (Table 4). These results concur with previous reports for balsam fir and black spruce [24,26].

Based on these results, the tendency for a weak negative relationship between RD and RW in mature wood may be explained by the fact that EWD, which was the most important parameter in determining wood density in mature wood, was not significantly correlated with RW (Table 4). However, this appears to be of no practical importance because the correlation coefficient was quite low (Table 4). Overall, this study suggests that a faster growth rate would not reduce wood density significantly in this species. Indeed, the negative relationships of RD and EWD with RW and EWW, although significant, were weaker (correlation coefficients varied from −0.23 to −0.10) in *T. occidentalis* compared to other species such as black spruce, where the correlation coefficients varied from −0.67 to −0.33 [27]. The same holds true for LP, however, the growth rate seemed to positively influence LWD

in this species (Table 4). These results contradict those found for balsam fir and black spruce, where a negative relationship was reported between LWD and RW [24,26].

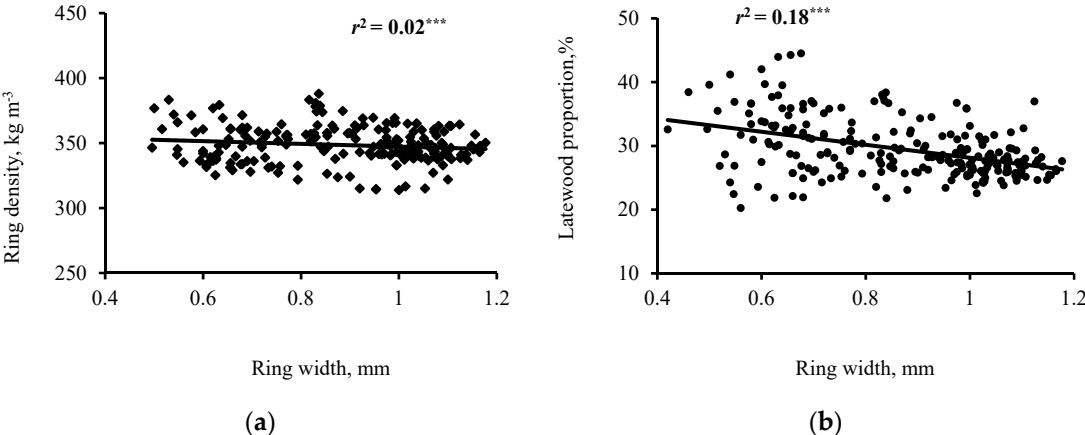

**Figure 4.** Variation of mean ring density (**a**) and latewood proportion (**b**) with mean ring width for 44 *T. occidentalis* trees. Significance level: *** $p < 0.001$.

### 3.3. Relationships between Ring Width and Tracheid Length and Width

The correlation coefficients between tracheid length and width, RW, tree diameter and height in *T. occidentalis* are shown in Table 6. Tracheid length correlated positively with tracheid width, tree diameter and tree height. However, tracheid length correlated negatively with RW. Tracheid width was also positively correlated with tree diameter and RW, but no relationship was found between tracheid width and tree height. Tree diameter was also positively and strongly correlated to tree height and RW (Table 6). These correlations are statistically significant at $p = 95\%$ except the tracheid width-tree height correlation (Table 6).

Several contradictory reports have been published on the relationship between tracheid length and RW in conifers and hardwoods [61–63]. An inverse relationship between tracheid length and RW in conifers was observed by Chalk [62], which is in good agreement with the results of the present study. Fujiwara and Yang [63] also reported a negative relationship between tracheid length and RW for jack pine, balsam fir and black spruce, but no relationship was found for white spruce. However, a positive relationship between tracheid length and RW was reported for trembling aspen [63]. Diaz-Váz et al. [64] also observed a positive relationship between tracheid length and RW in conifers. Dutilleul et al. [65] reported that fast-growing spruces (*Picea abies* (L.) Karst.) showed a stronger negative correlation between RW and tracheid length ($r = −0.86$).

The negative relationship between tracheid length and RW (Table 6) suggested that tracheid length depends on growth rate. According to Fujiwara and Yang [63], a tree's diameter growth is accompanied by the circumferential expansion of the cambium because the increase in cambium girth is primarily due to the increase in the number of fusiform initials achieved by pseudo-transverse division [16,55]. Bannan [55] considered the relationship between RW and cell length from the standpoint of RW and the pseudo-transverse division rate. The relationship between pseudo-transverse division rate and cell length is usually negative: a high rate is accompanied by short cells and, conversely, a low rate is accompanied by longer cells. Bannan [16] also showed that the frequency of pseudo-transverse division in fusiform initials is related to the linear radial increment in *T. occidentalis*. However, cell length is not only affected by RW, but also by circumferential growth. The circumferential growth rate differs according to tree diameter even though the radial growth rate is the same. Fujiwara and Yang [63] reported a negative relationship between tracheid length and circumferential growth rate in jack pine, balsam fir, black spruce and trembling aspen. In trembling aspen, tracheid length decreased with both higher and lower circumferential growth rates.

The negative correlations between RD and RW ($r = -0.10$ to $-0.19$) and between tracheid length and RW ($r = -0.12$) were weak in *T. occidentalis* compared to other species, such as jack pine, balsam fir, black spruce, trembling aspen and brutia pine (*Pinus brutia*, Ten.) [63,66], thus providing the opportunity for a silviculture program to simultaneously improve RW, wood density and tracheid length.

**Table 6.** Correlation coefficients between tracheid length, tracheid width, ring width, tree diameter and tree height in *T. occidentalis* and their statistical significance.

| Characteristic | Tracheid Length | Tracheid Width | Ring Width | Tree Height | Tree Diameter |
|---|---|---|---|---|---|
| Tracheid length | 1 | | | | |
| Tracheid width | 0.59 ** | 1 | | | |
| Ring width | −0.12 ** | 0.11 ** | 1 | | |
| Tree height | 0.11 ** | 0.06 ns | −0.02 ns | 1 | |
| Tree diameter | 0.15 ** | 0.33 ** | 0.12 ** | 0.54 ** | 1 |

Significance level: ** = $p < 0.01$ and ns = not significant.

## 4. Conclusions

*T. occidentalis* wood density is characterized by a slight increase from early- to latewood, which increases its value for use in wooden structures that require wood uniformity, such as veneer peeling and slicing. The latewood zone was narrow with thick cell walls and slightly denser than the earlywood zone. LP was slightly higher in juvenile than in mature wood. The intra-ring wood density variation was significantly greater in juvenile than in mature wood. Tracheids become smaller from early- to latewood. The cell and lumen areas decreased, whereas cell wall area increased. Cell and lumen diameter changes along a ring were greater in the radial direction than in the tangential one. By contrast, cell walls were thicker in the tangential direction compared to the radial direction.

The correlations between ring density components and ring widths were weaker in mature compared to juvenile wood. EWD seemed more important in determining overall wood density than LWD and LP. The relationship between RW and RD, although significant, was low and tended to weaken slightly with increasing tree age. This suggests that the negative impact of high growth on density decreases when the wood reaches maturity. EWW was strongly correlated to RW. However, LP and tracheid length were negatively correlated to RW. Accordingly, increases in RW produce wider earlywood without a corresponding increase in latewood.

**Author Contributions:** This study was part of a major research project where the principal investigator was A.K. and the co-investigator was Y.B. B.B. performed the wood quality experiments, the statistical analysis, analyzed the data, and wrote the manuscript as a part of her Ph.D. thesis under the supervision of A.K. and Y.B. All authors contributed in the conception and the design of the experiments and to the final manuscript writing.

**Acknowledgments:** The authors thank the Canada Research Chair Program, the Ministère des Forêts, de la Faune et des Parcs, Quebec (MRNF), and the NSERC-UQAT-UQAM Industrial Chair in Sustainable Forest Management for funding this project. We are also grateful to three thesis jury reviewers (Alain Cloutier, Timothy Work, and Hassine Bouafif) for their comments on an earlier draft of the manuscript.

**Conflicts of Interest:** The authors declare no conflict of interest.

## Appendix A

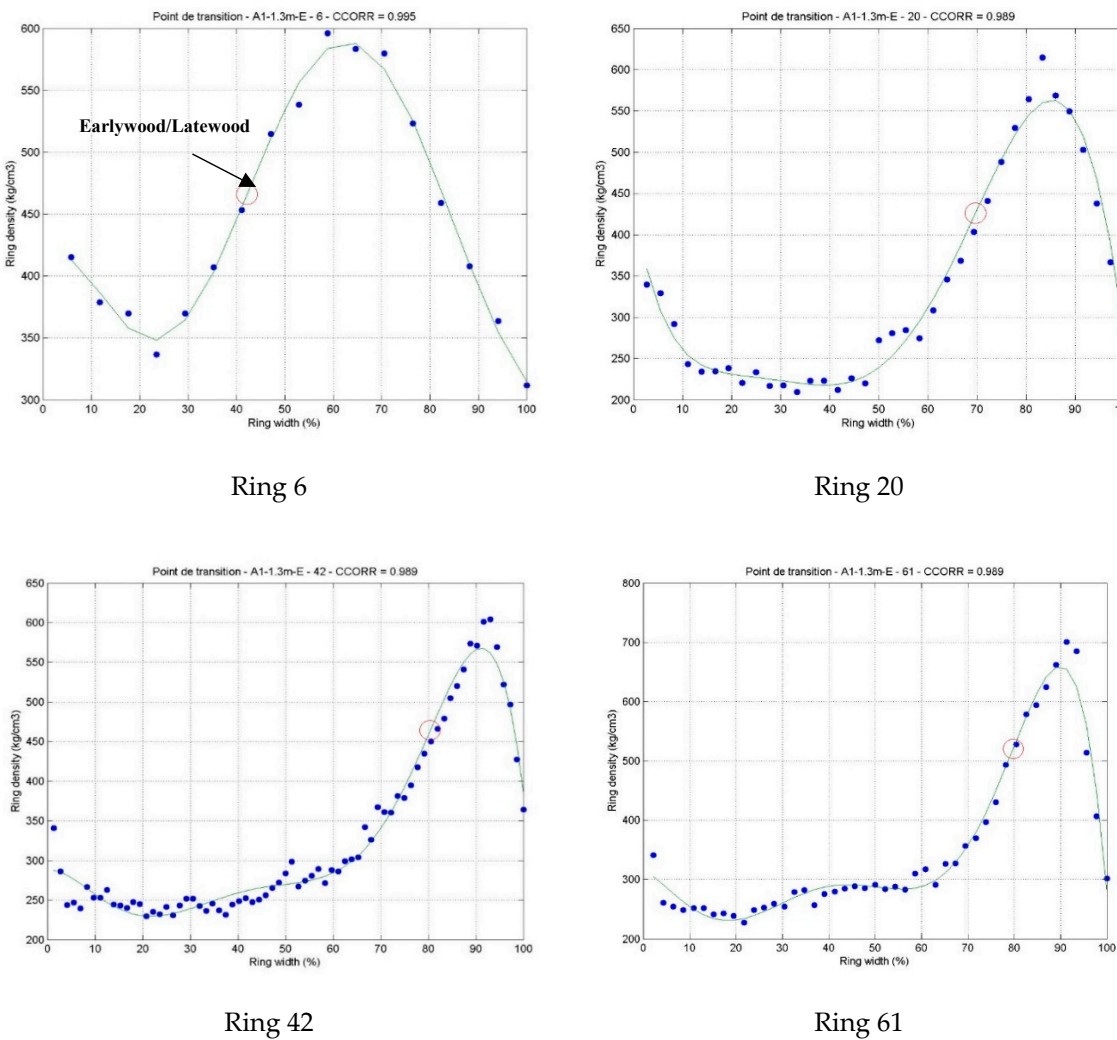

| Ring 6 | Ring 20 |
|:---:|:---:|

| Ring 42 | Ring 61 |
|:---:|:---:|

**Figure A1.** Example of intra- ring wood density profile modeled by Matlab software, using the maximum derivative method with a six-degree polynomial at selected growth rings.

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
