# Peer review of "Intra-Ring Variations and Interrelationships for Selected Wood Anatomical and Physical Properties of Thuja Occidentalis L."

_forests, doi:10.3390/f10040339_

Round 1

Reviewer 1 Report

The work and characterization of T. occidentalis wood is extensive adding knowledge and allowing suggestions for silvicultural practices. However the manuscript should be shortened for resubmission taking alsointo accountthat previous works were already publishedand some arguments and structure could be very similar. Besides, I would consider to change the title, and even if I am not a natural English speaker I would recommend a major revision of the English style and grammar since I noted some lack of verbs, inconsistent verb tenses, and misleading sentences. Moreover reading theprevious reports ofthe same authors thismanuscriptlacks for a careful attention regarding English, style and studieddata. Sentences such as those found in lines 102-103, lines 118-120 and 127-130 show some “errors” when compared to Bouslimi et al 2014 (reference 6), for example. Moreover, repeated meaning orexpressions should be avoided or rewritten, specially in Material and methods and also in results and Results and Discussion section such as “veneer and peeling” in lines 283-287.Besides there aremany misprints and mistakes that mightmislead alltherevision and thescientific findings. The abstract should be rewritten descriptions and facts that should be placed in other sections, for example the first 3 sentences and study aims should not be presented here. The same goes for Conclusions section.

I hope the following suggestions may help to improve the manuscript.

General and specific comments:

- EWC should be refereed at the beginning of the manuscript as it was in previous works or not given at all if it is author’s intention to use it;

- Variable and specific terms should be consistent along the manuscript,for example do the authors prefer growth ring, annual ring or tree ring? What is more adequate according for this manuscript and journal? What about intra-ring wood density density variation, within-ring andwithin growth rings? Also, why using “fibre length and width” to describe tracheids biometry and tracheid “properties” and so on? Consider also to use early- and latewood to make it more readable.

-Aparagraph and references on juvenile and mature wood of T. occidentalis related to stem and radial variation concepts and definitions should be presented in introduction. Tree age data (mean and range) by site/stand should be given in the Study material section;    

- Scientific names should be fullgiven when first mentioned as well as written in italic and not abbreviated in the beginning of the sentences would also be better;

- Abbreviated symbols should be corrected such as DBH instead of DHP and only presented once i.e. for the first appearance and not repeated as it happens in section 3.2 for instance;

- Correlation values refereed in the text should be consistent with values presented within the Tables, decimals should not be missing, and not repeated so often in both text and tables;

- Tables should present and specify the average and statistical values accordingly standards as well as avoid superscript numbers next to letters as seen in Table 1 (try to use superscript letters instead);

- Statistical terms such as mean and average, singular and plural should be reviewed as well as the variable trend adjectives used;

-Information about all theequipment must be given, for example information forthe sliding microtome is incomplete;

- Variable singular or plural terms should be revised and consistent with thisscientific area and journal;

- If wood density always refers to mean ring wood density it should be mentioned in the first place within the proper manuscript section. Do the authors prefer ring density or wood density (please see comment about terms preference and text uniformity)?

- Many sentences such as those in lines 375 – 377 could be rewritten as a singleone clarifying the scientific meaning;

-Since ring groups were created it is necessary an explanation about how and why they were created;

- Figure 3b and 3c are the same published in previous work (reference 28);

- Figure 3should allow readers to see structural differences between the juvenile and mature wood as discussed by the authors;

- Uniform microscopicalterms(please see comment about terms preference and text uniformity);

-Was LWD almost constant near the pith? Check it and compare withFigure 4;

- Table titles, variable descriptions or notes should be checked since terms might not be the most correct.

- Correlation analysis description should include the name of the used correlation as it was done for theother tests as well as insert itin table titlesor notes;

- Please clarify in the Statistical section if all procedures were performed using SAS and addinformation about the version and other software data;

- Is wood density variation in early- and latewood “a consequence” of the wood anatomical and chemical differences!?

- The juvenile and mature wood are the “nature” of the wood!?

- What dothe authors meanby differences in appearance and structure seen in the Figure 3!? Explain it, please.

- The coefficient of determination between RW and RD was very low but highly significant but no great explanation is discussed or given to this fact;

- The fact that all ring-porous hardwoods and some conifers and have latewood that is significantly more dense than the earlywood should be mentioned and compared in the discussion with references as well as the their wood density and latewood width pith-to-bark tendency;

- Abstract and Conclusions should be rewritten andshortened to avoid misleading section’s arguments or quantitative data repetition;

-If possible Section names should be revised and changed allowing an easier reading;

-Referencesneed to be checkedalong the text to confirm if all were inserted when author’s names are referred;

- The Acknowledgements section should specify the project name and funding as well as to givethe full organization/institute names. Moreover, information about any fellowships should be also given.

Author Response

Reviewer 1

ü     The work and characterization of T. occidentalis wood is extensive adding knowledge and allowing suggestions for silvicultural practices. However, the manuscript should be shortened for resubmission taking also into account that previous works were already published and some arguments and structure could be very similar. Besides, I would consider to change the title, and even if I am not a natural English speaker I would recommend a major revision of the English style and grammar since I noted some lack of verbs, inconsistent verb tenses, and misleading sentences. Moreover reading the previous reports of the same authors this manuscript lacks for a careful attention regarding English, style and studied data. Sentences such as those found in lines 102-103, lines 118-120 and 127-130 show some “errors” when compared to Bouslimi et al 2014 (reference 6), for example. Moreover, repeated meaning or expressions should be avoided or rewritten, specially in Material and methods and also in results and Results and Discussion section such as “veneer and peeling” in lines 283-287.Besides there are many misprints and mistakes that might mislead all the revision and the scientific findings. The abstract should be rewritten descriptions and facts that should be placed in other sections, for example the first 3 sentences and study aims should not be presented here. The same goes for Conclusions section

I hope the following suggestions may help to improve the manuscript

The authors addressed all comments in the revised version, shortened the manuscript and removed information that has been previously published. The title of the article and section names are changed. A professional English-speaking editor reviewed this version of the manuscript. The abstract and conclusions sections are rewritten according to the reviewers' comments.

-New title: Intra-Ring Variations and Interrelationships for Selected Wood Anatomical and Physical Properties of Thuja Occidentalis L.

General and specific comments:

ü     EWC should be refereed at the beginning of the manuscript as it was in previous works or not given at all if it is author’s intention to use it

-The word EWC is deleted from the manuscript (lines 46 and 101)

ü     Variable and specific terms should be consistent along the manuscript, for example do the authors prefer growth ring, annual ring or tree ring? What is more adequate according for this manuscript and journal? What about intra-ring wood density variation, within-ring and within growth rings? Also, why using “fibre length and width” to describe tracheids biometry and tracheid “properties (length and width)” and so on? Consider also to use early- and latewood to make it more readable.

-Addressed as requested

-Growth ring and annual ring are replaced by Tree ring (lines 15, 242, 250, 262, 442)

-Within-ring and within growth rings are replaced by intra-ring (lines 75, 79, 238, 325, 329, 364, 366, 373, and 595).

-To describe tracheids biometry and tracheid properties, fibre length and width are replaced by Tracheid length and width (lines 90, 169, 178 and 196)

ü     A paragraph and references on juvenile and mature wood of T. occidentalis related to stem and radial variation concepts and definitions should be presented in introduction. Tree age data (mean and range) by site/stand should be given in the Study material section;

-                      A paragraph and references on juvenile and mature wood of T. occidentalis related to stem and radial variation concepts and definitions are added (Lines 83-90).

«JW is one of the most important sources of inter- and intra-tree wood variation, particularly in conifers [21, 23]. JW forms a central core around the pith from tree base to top, following the crown as it grows [9]. Typically, the properties of JW make a gradual transition toward those of MW [9,24]. The radial variation in wood ring density and width and tracheid size of T. occidentalis is greater in JW than in MW [6]. Compared to MW, JW is composed of smaller, shorter tracheids, higher wood ring density, and latewood proportion, but lower ring width [6]. The longitudinal variation is more important in MW compared to JW [6]. Ring density and with decrease steadily from the tree base upward; however, tracheid length and width increase with increasing tree height [6]».

- Tree age data (mean and range) by site/stand are added in the Study material section (Lines 126-129)

«The age of sampled trees ranged from 60 to 198 years, with an average of 96 (60-134), 121 (73-198), and 93 (75-127) years for the Abitibi, Lac Duparquet and Témiscamingue sites, respectively. From each felled tree, 10-cm-thick disks were sampled at DBH».

ü      Scientific names should be full given when first mentioned as well as written in italic and not abbreviated in the beginning of the sentences would also be better;

- Addressed as requested

ü     Abbreviated symbols should be corrected such as DBH instead of DHP and only presented once i.e. for the first appearance and not repeated as it happens in section 3.2 for instance;

- Addressed as requested: DHP is replaced by DBH and not repeated in the text (Lines 124-130)

ü     7.            Correlation values refereed in the text should be consistent with values presented within the Tables; decimals should not be missing, and not repeated so often in both text and tables;

- Addressed as requested (sections 3.2 and 3.3).

ü     Tables should present and specify the average and statistical values accordingly standards as well as avoid superscript numbers next to letters as seen in Table 1 (try to use superscript letters instead (indice );

-Addressed as requested

Lines 265-271

Lines348-351

Lines 449-454

Lines 495-500

ü     Statistical terms such as mean and average, singular and plural should be reviewed as well as the variable trend adjectives used;

-Addressed as requested: average is replaced by mean

A professional English-speaking editor reviewed this version of the manuscript

ü     Information about all the equipment must be given, for example information for the sliding microtome is incomplete;

- Information about all the equipment is added (lines 139-140; 179-180 and 186-187)

- QTRS-01X Tree-Ring X-Ray Scanner (Quintek Measurement Systems, Knoxville, Tennessee, USA)

- Fiber Quality Analyzer (FQA, OPTEST Equipment Inc., Hawkesbury, Ontario, Canada)

-sliding microtome (HM 325 Rotary Microtome, Kalamazoo, USA)

ü     Variable singular or plural terms should be revised and consistent with this scientific area and journal

- A professional English-speaking editor reviewed this version of the manuscript

ü     If wood density always refers to mean ring wood density it should be mentioned in the first place within the proper manuscript section. Do the authors prefer ring density or wood density (please see comment about terms preference and text uniformity)?

-Wood density is replaced by Ring density (sections 2.2 and 2.3)

On the other hand, when talking about intra-ring variation (section 3.1), we used the most common term to describe the intra-ring wood density variation.

i.e:

Decoux, Valérie, Éliane Varcin et Jean-Michel Leban. 2004. «Relationships between the intra-ring wood density assessed by X-ray densitometry and optical anatomical measurements in conifers. Consequences for the cell wall apparent density determination». Annals of Forest Science,  vol. 61, no 3, p. 251-262.   

Rathgeber, Cyrille BK, Valérie Decoux et Jean-Michel Leban. 2006. «Linking intra-tree-ring wood density variations and tracheid anatomical characteristics in Douglas fir (Pseudotsuga menziesii (Mirb.) Franco)». Annals of Forest Science,  vol. 63, no 7, p. 699-706.

ü     Many sentences such as those in lines 375 – 377 could be rewritten as a single one clarifying the scientific meaning; reformer.

-Addressed as requested: the sentence

«The patterns of within-ring variation were relatively similar between rings from juvenile and mature wood (Figure 1). However, EWD in the juvenile wood zone was slightly higher than that in the mature wood. Conversely, LWD in juvenile wood was slightly lower than that in mature wood. Furthermore, the within-ring distribution of latewood was slightly more important in juvenile (32-40%) than in mature (20%) wood (Figure 1)» is replaced by  (Lines 322-326) «The patterns of intra-ring variation were relatively similar between rings from JW and MW (Figure 1). However, the intra-ring distribution of latewood was slightly more important in JW (32-40%) than in MW (20%) (Figure 1)».

ü     Since ring groups were created it is necessary an explanation about how and why they were created;

-Addressed as requested (lines 163-168)

«Matlab software was used to model the intra-ring wood density profiles, using the maximum derivative method [21]. The intra-ring wood density variation was calculated for whole tree and JW (rings 2-30) and MW (rings 31-100) (Figure 1). The intra-ring wood density variation was also considered for selected ring groups (rings 2-10, rings 11-20 and every 10th annual ring up to the bark) (Figure 2) to investigate the cambial age effect on this variation.»

ü     Figure 3b and 3c are the same published in previous work (reference 28).

-Figures 3b and 3c are deleted and two new figures are added. The order of Figure 3 has become Figure 2 in the revised version.

-The text is adapted to figure 2 (lines 247-264)

«The intra-ring wood density (Figure 1) and tracheid size variations (Figure 2a and Table 1) revealed that the transition from early- to latewood is gradual in T. occidentalis. The early- and latewood were distinguished in the structure (Figure 2 and Table 1). The latewood zone was narrow with thick cell walls (Figure 2b, Table 1) and was slightly denser than the earlywood zone (Figure 1). Thuja occidentalis wood was relatively homogeneous and simple in structure, consisting primarily of overlapping tracheids (between 23.58 and 26.19 µm) connected by uniseriate xylem rays, parenchyma cells (Figure 2b) and bordered pits, which were visible and abundant on the radial face (Figures 2c and d). The tracheid cell walls were organized in layers of different thicknesses: thick latewood cell walls and thin earlywood cell walls (Figure 2b, Table 1). In earlywood, the longitudinal tracheids were hexagonal with minimal wall thickness and a larger diameter, usually in the radial direction, while the diameter of longitudinal tracheids in the tangential direction remained relatively constant within a tree ring (Figure 2b). In latewood, the cross sections of tracheids were essentially rectangular and compacted radially to a tabular shape (Figure 2b).»

ü     Figure 3 should allow readers to see structural differences between the juvenile and mature wood as discussed by the authors;

-The following sentence is added according to figure 3 (Lines 364-366)

« The intra-ring wood density distribution was more homogeneous in MW than in JW (Figure 3, Table 2)».

ü     Uniform microscopical terms (please see comment about terms preference and text uniformity);

- Addressed as requested: In the revised version, we used the term anatomical properties to describe microscopic analysis.

ü     Was LWD almost constant near the pith? Check it and compare with Figure 4;

-LWD was low near the pith. The sentence is corrected in revised version according to figure 4 (Lines 465-466).

«LWD was low near the pith, increased to a maximum in the juvenile-mature transition zone (30 years) and remained constant thereafter».

ü Table titles, variable descriptions or notes should be checked since terms might not be the most correct.

- Addressed as requested

-See Lines 265-271

-Lines 340-343

-Lines348-351.

-Lines 449-454

ü     Correlation analysis description should include the name of the used correlation as it was done for the other tests as well as insert it in table titles or notes;

-Correlation analysis description is added in the text and table titles

-Lines 229-230. «Pearson’s correlation coefficients were also computed using the CORR procedure to determine relationships between all wood density components in JW and MW, and for selected ring groups»

Lines 445-450: Table 4. Pearson correlation coefficients between ring density components and ring widths in juvenile and mature wood of T. occidentalis and their statistical significance.

-Lines 497-498: Table 5. Pearson correlation coefficients between ring density components and ring widths for selected ring groups in T. occidentalis and their statistical significance.

ü     Please clarify in the Statistical section if all procedures were performed using SAS and add information about the version and other software data;

- Addressed as requested:

-Lines 234-235. «All statistical and correlations analyses were performed using SAS® (SAS Institute Inc., Cary, NC, USA) [37] ».

ü     Is wood density variation in early- and latewood “a consequence” of the wood anatomical and chemical differences!?

-Variation in early- and latewood is principally a consequence of the wood anatomical modifications. The sentence is corrected:

-Lines 75-77: The sentence« It also varies greatly from earlywood to latewood as a consequence of anatomical and chemical modifications [10, 18, 50] » is replaced by « The variation in intra-ring wood density is related to cambial activity and varies with age [18, 22]. It also varies greatly from early- to latewood because of anatomical modifications [10, 18]s».

-Line 386-388. «According to several works, the anatomical changes seem to be the main responsible for intra-ring wood density variation [10, 18, 49] ».

ü     The juvenile and mature wood are the “nature” of the wood!?

-The juvenile and mature wood are the types of wood

-Lines 77-78: The sentence «The within-ring variation depends on species and the nature of the wood (juvenile and mature wood) [21, 23]». Is replaced by «The intra-ring variation depends on species and the type of the wood: juvenile wood (JW) and mature wood (MW) [21, 23]».

ü     What do the authors mean by differences in appearance and structure seen in the Figure 3!? Explain it, please.

- It is a mistake, we want to say the early- and latewood were distinguished in the structure

-Lines 248-252. The sentence «The earlywood and latewood differ in appearance and structure (Figure 3). Within a growth ring, the latewood has a dark color that differentiates it from the earlywood (Figure 3a). Furthermore, the latewood zone is narrow with thick cell walls (Figure 3c, Table 1) and is slightly denser than the earlywood zone (Figure 1). » is replaced by «The early- and latewood were distinguished in the structure (Figure 2 and Table 1). The latewood zone was narrow with thick cell walls (Figure 2b, Table 1) and was slightly denser than the earlywood zone ».

ü     The coefficient of determination between RW and RD was very low but highly significant but no great explanation is discussed or given to this fact;

-An explanation is added:

-Lines 484-496 «The coefficient of determination between RW and RD was low but significant. According to Koga and Zhang [57], a slow increase in RW could negatively affect RD. The percentage of early-and latewood in a tree ring determines the overall density of the ring, and in T. occidentalis, the LWW is relatively constant. Therefore, the RW is closely related to the EWW (Figure 4). As the RW increases, the width of the earlywood increases without a corresponding increase for latewood, causing lower RD [6]. This could explain the non-significant correlation between RD and LWW in both JW and MW, as well as the negative relationship of RD with LP (Tables 4 and 5). In JW, earlywood cells have a thin wall layer (Table 1), which is one of the reasons for lower density [9]. According to Zhang et al. [60], the decrease in RD combined with the increase in RW in conifers was more pronounced in species that showed a gradual transition from early- to latewood (as T. occidentalis) than in species that made an abrupt transition.»

ü     The fact that all ring-porous hardwoods and some conifers and have latewood that is significantly more dense than the earlywood should be mentioned and compared in the discussion with references as well as the their wood density and latewood width pith-to-bark tendency;

-Information is added for ring porous-hardwoods

-Lines 409-412. «For hardwoods such as green ash (Fraxinus pennsylvanica Marsh) [53] and red maple (Acer rubrum L.) [54], the higher LWD was attributed to an increase in fiber cell wall thickness and a decrease in vessel diameter compared to those of earlywood».

ü     Abstract and Conclusions should be rewritten and shortened to avoid misleading section’s arguments or quantitative data repetition;

-Addressed as requested

Abstract: Intra-ring variation in wood density and tracheid anatomical properties and wood property interrelationships were investigated in Thuja occidentalis L. Samples were taken from three stands in Abitibi-Témiscamingue, Quebec, Canada. The structure of T. occidentalis wood is simple, homogeneous and uniform, which is desirable for wooden structures that require wood uniformity. From early- to latewood, cell and lumen diameter decreased, while cell wall thickness increased. These changes led to an increase in the cell wall proportion. Wood ring density and width interrelationships were weaker in mature wood compared to juvenile wood. Earlywood density (r = 0.92) is the most important in determining mature wood density than latewood density (r = 0.53) and proportion (r = 0.23). Earlywood density explains 97 and 79 % (r2) of the variation in juvenile and mature wood density, respectively. The negative relationship between ring density and width, although significant, was low (r <-0.1) and tends to weaken slightly with increasing tree age, thus providing the opportunity for silvicultural practices to simultaneously improve growth and wood density. Ring width was positively and strongly correlated to early- and latewood width, but negatively correlated to tracheid length and latewood proportion. Accordingly, increases in ring width produce smaller tracheids and wider earlywood without a corresponding increase in latewood. Practical implications for end-uses are discussed.

Conclusions: Thuja occidentalis wood density is characterized by a slight increase from early- to latewood, which increases its value for use in wooden structures that require wood uniformity, such as veneer peeling and slicing. The latewood zone was narrow with thick cell walls and slightly denser than the earlywood zone. LP was slightly higher in JW than in MW. The intra-ring wood density variation was significantly greater in JW than in MW. Tracheids become smaller from early- to latewood. The cell and lumen areas decreased, whereas cell wall area increased. Cell and lumen diameter changes along a ring were greater in the radial direction than in the tangential one. By contrast, cell walls were thicker in the tangential direction compared to the radial direction. The correlations between ring density components and ring widths were weaker in MW compared to JW. EWD seemed more important in determining overall wood density than LWD and LP. The relationship between RW and RD, although significant, was low and tended to weaken slightly with increasing tree age. This suggests that the negative impact of high growth on density decreases when the wood reaches maturity. EWW was strongly correlated to RW. However, LP and tracheid length were negatively correlated to RW. Accordingly, increases in RW produce wider earlywood without a corresponding increase in latewood.

ü     If possible Section names should be revised and changed allowing an easier reading;

-Section names are changed:

2.2. Wood ring density and width measurement

2.3. Tracheid length and width measurement

2.4. Microscopic analysis of tracheid anatomical properties

3.1. Intra-ring wood density variation and anatomical changes between early-and latewood

3.3. Relationships between ring width and tracheid length and width

ü     References need to be checked along the text to confirm if all were inserted when author’s names are referred;

-References are checked along the text.

ü     The Acknowledgements section should specify the project name and funding as well as to give the full organization/institute names. Moreover, information about any fellowships should be also given.

-Information is added(Lines 617-621)

«Acknowledgments: The authors thank the Canada Research Chair Program, the Ministère des Forêts, de la Faune et des Parcs, Quebec (MRNF), and the NSERC-UQAT-UQAM Industrial Chair in Sustainable Forest Management for funding this project. We are also grateful to three thesis jury reviewers (Alain Cloutier, Timothy Work, and Hassine Bouafif) for their comments on an earlier draft of the manuscript.

Reviewer 2 Report

General comment

The manuscript presents interesting results on the characterization of the wood properties of the species Thuja occidentalis L. and its possible implications in wood quality. It is well written, with a detailed description of the measured wood properties.

Minor comments

Please check the order of Figures and Tables. For example, in page 5, line 199 it is mentioned Figure 2 but in the text appears after Figure 3. I understand this has a final editing procedure by the journal but when reading the submitted version it is much easier if the figures and tables follow a logical order in their position within the text.

Line 246: Acacia mangium in italics.

Line 305: “In earlywood, woody density decreased with cambial age.”

In Table 4 what do you mean by upper triangle and lower triangle?

Author Response

Reviewer 2

Comments and Suggestions for Authors

General comment

The manuscript presents interesting results on the characterization of the wood properties of the species Thuja occidentalis L. and its possible implications in wood quality. It is well written, with a detailed description of the measured wood properties.

Minor comments

ü     Please check the order of Figures and Tables. For example, in page 5, line 199 it is mentioned Figure 2 but in the text appears after Figure 3. I understand this has a final editing procedure by the journal but when reading the submitted version it is much easier if the figures and tables follow a logical order in their position within the text.

--Addressed as requested

ü     Line 246: Acacia mangium in italics.

- Acacia mangium is changed in italics

ü     Line 305: “In earlywood, woody density decreased with cambial age.”

-Addressed as requested (line 363)

ü     In Table 4 what do you mean by upper triangle and lower triangle?

-The title is changed (Lines 446-448)

-Table 4. Pearson correlation coefficients between ring density components and ring widths in juvenile and mature wood of T. occidentalis and their statistical significance.

Round 2

Reviewer 1 Report

Taking into consideration that previous recommendations were addressed I would like to leave just these final comments to improve the final manuscript.

Comments:

Abbreviations were in general carefully addressed as requested. However I would recommend not to use abbreviations for juvenile and mature wood since it is not common and from my point of view not necessary. In the case of earlywood abbreviation in parenthesis as “(early-/latewood)” it is confuse to read and should be changed. Even if I have only gave general directions to author’s, I would recommend to change “tree ring” and “tree-ring” terms usually used in dendrochronology to “growth ring” instead.

Figure 2 (a or b) must include one annual growth ring where earlywood transition to latewood is seen during the growth season, and not just the growth ring boundaries. This is crucial to confirm ring structure and the abrupt or gradual transition from earlywood to latewood as well as tracheid’s development within the growth ring as discussed in the text. Figure 2 c or 2 d should be deleted and added a new picture showing the uniseriate rays in the tangential section. Arrows seen in the picture should be mentioned within the figure caption in parenthesis after the respective anatomical character.

Anatomy description even if small must be rewritten “consisting primarily of overlapping tracheids (between 23.58 and 26.19 μm), connected by uniseriate xylem rays, parenchyma cells (Figure 2b, Table 1) and bordered pits, which were visible and abundant on the radial face (Figures 2c and d).” When the authors refer “between 23.58 and 26.19 μm” what do they mean? This should be cleared and also the wood direction of the referred measurements. Are tracheids always connected by uniseriate xylem rays? This not seen in the Figure so rephrased it for example as “uniseriate xylem rays” also according the previous published work (ref. 7). The authors must specify axial or ray parenchyma cells and not only “parenchyma cells” since as known the axial parenchyma cells are a valuable character for softwood species which must be seen in the cross section. Please verify if this is the case of your samples and include a cross section showing the axial parenchyma of T. occidentalis. Did you find it diffuse or tangentially zonate? Please mention and show it in a new image (see previous comment). Tracheid bordered pits are observed in radial section so “on the radial face” expression should be deleted.

Author Response

Reviewer 1

ü    Abbreviations were in general carefully addressed as requested. However, I would recommend not to use abbreviations for juvenile and mature wood since it is not common and from my point of view not necessary. In the case of earlywood abbreviation in parenthesis as “(early-/latewood)” it is confuse to read and should be changed. Even if I have only gave general directions to author’s, I would recommend to change “tree ring” and “tree-ring” terms usually used in dendrochronology to “growth ring” instead.

Addressed as requested

-JW is replaced by juvenile wood (lines 62-81, 128-204,278-319, 362-444 and 499-506)

-MW is replaced by mature wood (lines 63-82, 129-136,197-319, 362-372, 386-399, 402-417, 427-447 and 499-506)

-Tree ring is replaced by growth ring (lines 185, 261, 267, 338-339 and 345)

-(early-/latewood) is replaced by (earlywood/latewood) (Lines 44 and 282)

ü    Figure 2 (a or b) must include one annual growth ring where earlywood transition to latewood is seen during the growth season, and not just the growth ring boundaries. This is crucial to confirm ring structure and the abrupt or gradual transition from earlywood to latewood as well as tracheid’s development within the growth ring as discussed in the text. Figure 2 c or 2 d should be deleted and added a new picture showing the uniseriate rays in the tangential section. Arrows seen in the picture should be mentioned within the figure caption in parenthesis after the respective anatomical character.

Addressed as requested

-Figures 2a and 2b are replaced by two new figures where earlywood to latewood transition is seen during the growth season (Figures a and b). The uniseriate rays is difficult to see in the tangential section but it is very clear in the Transverse section (figure 2c).On the other hand, in the tangential section, uniseriate or rarely occasionally biseriate  bordered pits were observed, which are a characteristics of this wood (Figure 2d).

-The text is adapted to figure 2 (lines 211-226)

« The intra-ring wood density (Figure 1) and tracheid size variations (Figure 2 and Table 1) revealed that the transition from early- to latewood is gradual in T. occidentalis. The early- and latewood were distinguished in the structure (Figure 2 and Table 1). The latewood zone was narrow with thick cell walls (Figures 2a, b and c, Table 1) and was slightly denser than the earlywood (Figure 1a). Thuja occidentalis wood was relatively homogeneous and simple in structure, consisting primarily of overlapping tracheids (tracheids average 23.58 µm and 26.19 µm in the tangential and in the radial diameter, respectively) connected by as uniseriate xylem rays, ray parenchyma cells (Figure 2c, Table 1) and uniseriate or occasionally biseriate  bordered pits (Figure 2d). Axial parenchyma is usually rare or absent. Rays are uniseriate and thin (Figure 2c).The tracheid cell walls were organized in layers of different thicknesses: thick latewood cell walls and thin earlywood cell walls (Figures 2a, b and c, Table 1). In earlywood, the longitudinal tracheids were hexagonal with minimal wall thickness and a larger diameter, usually in the radial direction, while the diameter of longitudinal tracheids in the tangential direction remained relatively constant within a growth ring (Figures 2a and c). In latewood, the cross sections of tracheids were essentially rectangular and compacted radially to a tabular shape (Figures 2a and c)»

ü    Anatomy description even if small must be rewritten “consisting primarily of overlapping tracheids (between 23.58 and 26.19 μm), connected by uniseriate xylem rays, parenchyma cells (Figure 2b, Table 1) and bordered pits, which were visible and abundant on the radial face (Figures 2c and d).” When the authors refer “between 23.58 and 26.19 μm” what do they mean? This should be cleared and also the wood direction of the referred measurements. Are tracheids always connected by uniseriate xylem rays? This not seen in the Figure so rephrased it for example as “uniseriate xylem rays” also according the previous published work (ref. 7). The authors must specify axial or ray parenchyma cells and not only “parenchyma cells” since as known the axial parenchyma cells are a valuable character for softwood species which must be seen in the cross section. Please verify if this is the case of your samples and include a cross section showing the axial parenchyma of T. occidentalis. Did you find it diffuse or tangentially zonate? Please mention and show it in a new image (see previous comment). Tracheid bordered pits are observed in radial section so “on the radial face” expression should be deleted.

-Addressed as requested (Lines 215-220)

-Tracheids are not always connected by uniseriate xylem rays

- Axial parenchyma is usually rare or absent in this species.

-on the radial face” expression is deleted

-The sentence

«Thuja occidentalis wood was relatively homogeneous and simple in structure, consisting primarily of overlapping tracheids (between 23.58 and 26.19 µm) connected by uniseriate xylem rays, parenchyma cells (Figure 2b, Table 1) and bordered pits, which were visible and abundant on the radial face (Figures 2c and d).»

is replaced by ((Lines 215-220)

«Thuja occidentalis wood is relatively homogeneous and simple in structure, consisting primarily of overlapping tracheids (tracheids average 23.58 µm and 26.19 µm in tangential and in the radial diameter, respectively) connected by uniseriate xylem rays, ray parenchyma cells (Figure 2c, Table 1) and uniseriate or rarely occasionally biseriate  bordered pits (Figure 2d). Axial parenchyma is usually rare or absent. Rays are uniseriate and thin (Figure 2c))»